# LoReUn: Data Itself Implicitly Provides Cues to Improve Machine Unlearning

**Xiang Li**    **Qianli Shen**    **Haonan Wang**    **Kenji Kawaguchi**

School of Computing
National University of Singapore
{xiang_li,shenqianli,haonan.wang}@u.nus.edu, kenji@comp.nus.edu.sg

## Abstract

Recent generative models face significant risks of producing harmful content, which has highlighted machine unlearning (MU) as a crucial method for removing the influence of undesired data. However, different difficulty levels among data points can affect unlearning performance. In this paper, we propose that the loss of a data point implicitly reflects its varying difficulty level, leading to our plug-and-play strategy, Loss-based Reweghting Unlearning (LoReUn), which dynamically reweight data throughout the unlearning process with minimal computational effort. Our method significantly reduces the performance gap with exact unlearning in both image classification and generation tasks, effectively enhancing the prevention of harmful content generation from text-to-image diffusion models.

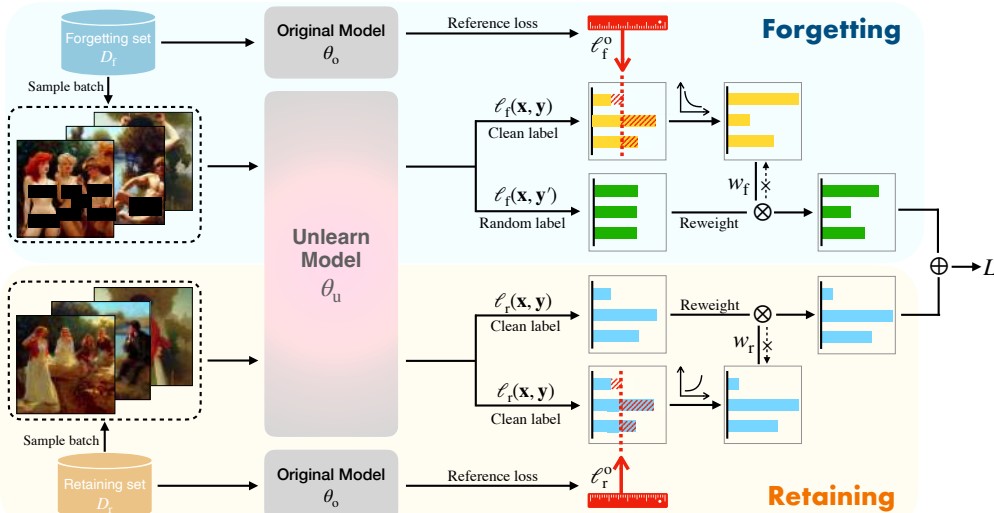

Figure 1: Given a forgetting set that contains data to be unlearned and a retaining set of remaining data from the training set, the unlearned model updates upon the original model to remove the influence of the forgetting data, while maintaining utility of the retaining data. To achieve this, the overall unlearning loss consists of two parts: forgetting loss, which uses random labels for the forgetting data, and retaining loss, which employs clean labels for the retaining data. We propose Loss-based Reweghting Unlearning (LoReUn) to dynamically reweight data based on the difference between the current loss on the unlearned model and a reference loss on the original model, enabling effective and efficient handling of data with varying difficulties during the forgetting and retaining process.

38th Conference on Neural Information Processing Systems (NeurIPS 2024).

# 1 Introduction

As generative models have grown rapidly in size and capacity, they unintentionally memorize sensitive, private, harmful, or copyrighted information from their training data [3, 25]. This causes the potential risk to generate inappropriate content when triggered by certain inputs. For instance, researchers have shown that text-to-image generative models are particularly prone to generating undesirable content, such as nudity or violence, when exposed to inappropriate prompts [21]. In response, machine unlearning (MU) has gained renewed attention as a strong strategy to eliminate the influence of specific data points for building trustworthy machine learning systems. Exact MU methods [11, 2], such as retraining from scratch without the forgetting dataset, offer provable unlearning guarantees but are computationally expensive, making them impractical for real-world usage. To this end, most works [15, 28, 10, 26, 5] focus on approximate MU methods to achieve a balance between unlearning effectiveness and efficiency. As an emerging area of research, approximate unlearning still has significant potential for improvement to narrow the performance gap with exact MU.

To understand the limitations and mechanisms behind existing approximate MU methods, several efforts have focused on analyzing data that is relatively challenging to unlearn. For example, Fan et al. [6] finds unlearning can fail when evaluated on the worst-case forget set. Barbulescu and Triantafillou [1] suggests treating data individually based on how well the original model memorizes it, while a following work [30] examines how entanglement and memorization degrees affect the unlearning difficulty of different data. However, the previous approaches are too computationally expensive to dynamically identify the difficulty of data points [30].

In this paper, instead of explicitly evaluating the difficulty of each data point, we present the idea that the data itself's loss can implicitly reflect its varying difficulty during unlearning. We introduce a simple yet effective plug-and-play strategy, **Lo**ss-based **Re**weighing for **Un**learning (LoReUn), which dynamically reweights data according to the current loss on the unlearned model and a reference loss from the original model. This reweighting process requires no additional inference for the data, making it significantly more lightweight compared to previous methods or exact MU. Our experimental results demonstrate that LoReUn remarkably reduces the performance gap with exact MU approaches, providing an effective and practical approach in both image classification and generation tasks. Notably, LoReUn excels in the application of eliminating harmful images generated from stable diffusion with inappropriate prompts (I2P [21]).

# 2 Related Work

## 2.1 Safety Concerns in Generation Models

Generative models like diffusion models are usually trained on data sets collected from diverse open sources, such as LAION [22]. This causes them to face the risk of generating inappropriate images [21] or copyright-infringed content by mimicking artistic style [24]. While [19] argue that simply using a safety filter can not prevent diffusion models from generating harmful content, another line of works [7, 17, 8, 21, 29, 5, 13] studies erasing unsafe concepts from pre-trained diffusion models to mitigate undesirable generations.

## 2.2 Machine Unlearning

Machine Unlearning (MU) aims to eliminate the influence of specific data points from a pre-trained model [9, 18, 27, 23]. Though retraining from scratch provides exact unlearning [2], it suffers from impractical computation demands. Alternatively, approximate unlearning methods [28, 10, 26, 15, 4] have been proposed as a more effective and efficient solution. Among them, Random Labeling [10] (RL) proposes to fine-tune the model with randomly labeled forgetting data. SalUn [5] further introduced gradient-based weight saliency to update specific model weights instead of the whole model during the unlearning process. The application of SalUn covers both image classification and generation tasks. In this paper, we design an effective plug-and-play strategy for RL-based unlearning methods.

# 3 Preliminaries and Problem Statement

**Machine Unlearning.** Let $\mathcal{D} = \{\mathbf{z}_i\}_1^N$ be the training set consisting of $N$ data points, where each data point is represented as feature $\mathbf{x}_i$ with or without label $y_i$. Let $\mathcal{D}_{\mathrm{f}}$ be the forgetting dataset and $\mathcal{D}_{\mathrm{r}} = \mathcal{D} \backslash \mathcal{D}_{\mathrm{f}}$ be the remaining dataset. The original model $\boldsymbol{\theta}_{\mathrm{o}}$ is trained on $\mathcal{D}$ and we regard Retrain as the gold standard model $\boldsymbol{\theta}$, which is trained on $\mathcal{D}_{\mathrm{r}}$ only from scratch. The problem of machine unlearning is to obtain an unlearned model $\boldsymbol{\theta}_{\mathrm{u}}$ from the original model $\boldsymbol{\theta}_{\mathrm{o}}$ using $\mathcal{D}_{\mathrm{f}}$ with or without $\mathcal{D}_{\mathrm{r}}$, such that it can be a perfect surrogate model for Retrain model $\boldsymbol{\theta}$ but is much more computationally efficient. One of the biggest challenges in MU is to balance between unlearning efficacy on $\mathcal{D}_{\mathrm{f}}$ and model utility on $\mathcal{D}_{\mathrm{r}}$ as well as generalization ability on test set $\mathcal{D}_{\mathrm{t}}$.

**Machine Unlearning for Classification.** There are two scenarios for machine unlearning in image classification: class-wise forgetting and random data forgetting. The former task aims to remove the influence of an image class, while the latter aims to forget a subset of randomly selected data points from the training set. In this paper, we will focus on one of the most effective MU method Random Labeling (RL). The objective of RL for unlearning in classification can be formulated as:

$$L^{(1)}(\boldsymbol{\theta}_{\mathrm{u}}) = \mathbb{E}_{(\mathbf{x},y)\sim\mathcal{D}_{\mathrm{f}}, y'\neq y}[\ell_{\mathrm{CE}}(\boldsymbol{\theta}_{\mathrm{u}}; \mathbf{x}, y')] + \alpha\mathbb{E}_{(\mathbf{x},y)\sim\mathcal{D}_{\mathrm{r}}}[\ell_{\mathrm{CE}}(\boldsymbol{\theta}_{\mathrm{u}}; \mathbf{x}, y)], \tag{1}$$

where $y'$ is the random label of $\mathbf{x}$ different from $y$, $\alpha > 0$ is a regularization parameter.

**Machine Unlearning for Generation.** In this paper, we focus on unlearning in conditional latent diffusion model Stable Diffusion [20]. Text-to-image diffusion models use prompts as conditions to guide the sampling process for generating images, which may contain unsafe content with inappropriate prompts as input. The training of diffusion models consists of a predefined forward process adding noise to data and a reverse process denoising the corrupted data, with its loss given by:

$$\ell_{\mathrm{SD}}(\boldsymbol{\theta}; \mathcal{D}) = \mathbb{E}_{t,(\mathbf{z},c)\sim\mathcal{D},\epsilon\sim\mathcal{N}(0,1)}\left[\|\epsilon - \epsilon_{\boldsymbol{\theta}}(\mathbf{z}_t|c)\|_2^2\right], \tag{2}$$

where $\mathbf{z}_t$ is a noisy latent of $\mathbf{z}$ at timestep $t$, $\epsilon_{\boldsymbol{\theta}}(\mathbf{z}_t|c)$ is the noise estimation given conditioned prompt $c$. Unlearning in image generation also encompasses a trade-off between two objectives: eliminating undesired content generated from the pre-trained diffusion model when conditioned on forgetting concepts like nudity, and preserving the quality of normal images generated from the unlearned model. Accordingly, following [5], the unlearning loss of random labeling in diffusion models becomes twofold:

$$L^{(2)}(\boldsymbol{\theta}_{\mathrm{u}}) = \mathbb{E}_{t,(\mathbf{z},c)\sim\mathcal{D}_{\mathrm{f}},\epsilon\sim\mathcal{N}(0,1),c'\neq c}\left[\|\epsilon_{\boldsymbol{\theta}_{\mathrm{u}}}(\mathbf{z}_t|c') - \epsilon_{\boldsymbol{\theta}_{\mathrm{u}}}(\mathbf{z}_t|c)\|_2^2\right] + \beta\ell_{\mathrm{SD}}(\boldsymbol{\theta}_{\mathrm{u}}; \mathcal{D}_{\mathrm{r}}), \tag{3}$$

where $c'$ is a concept different from $c$, $\beta > 0$ is a regularization parameter.

# 4 Method

In this section, we introduce a lightweight plug-and-play strategy to dynamically manage data of diverse difficulty levels during the unlearning process. The key idea of our method is to reweight data according to the current loss on the unlearned model and a reference loss from the original model.

**Motivation.** Previous works [6, 1, 30] have shown that the difficulty of different data points can result in varied unlearning performance. However, adding extra steps to estimate data difficulty can be computationally expensive during the unlearning process. This motivates our method of using the difference of current loss on the unlearned model and a reference loss from the original model to implicitly capture the dynamic difficulty of each data point according to the model's response over time. In Figure 2, we illustrate the average loss of the original and unlearned models on the three sets. We can observe that compared with the original model,

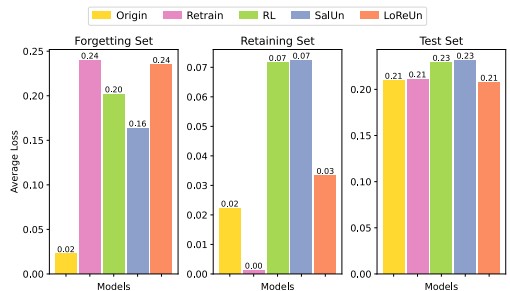

Figure 2: The average loss on different sets of data. LoReUn acheives the smallest gap with exact MU (Retrain) compared to baseline models RL and SalUn.

exact MU (Retrain) shows a significant increase in average loss on the forgetting set, while the average loss on the retaining set slightly decreases , and remains similar on the test set. This observation is consistent with the expectation that an ideal unlearned model should exhibit increased loss in the forgetting set and preserved loss performance in the retaining set and test set. It also implies that the unlearning efficacy and retaining ability of MU models can be implicitly reflected in the loss values.

Building on this motivation, we introduce a simple yet effective plug-and-play unlearning strategy to increase the weight of data points in the forgetting set if their loss decreases, while for the retaining set, we lower the weight of data points with dropped loss.

A detailed algorithm for our proposed LoReUn is provided in Algorithm 1.

**Step 1: Obtain reference losses with the original model**   We use a reference loss to provide a point of comparison for tracking the unlearning progress, as it would be hard to differentiate data of varying difficulties without an initial gauge. Specifically, we compute an averaged loss $\ell_{\mathrm{f}}^{\mathrm{o}}$ of forgetting dataset ($\ell_{\mathrm{r}}^{\mathrm{o}}$ for retaining dataset) on the original model $\boldsymbol{\theta}_{\mathrm{o}}$. For the image generation task, however, the magnitudes of loss values are not the same for data at different sampled time steps $t$ within a batch when training diffusion models (overall speaking, larger time steps result in smaller loss magnitudes). However, computing an initial average loss for all time steps is time-consuming. Instead, we first computed average loss at sampled time step $t$ with a fixed interval (e.g., a total of 10 time steps ranging from 0 to 1000 with an interval of 100). We then fit the sampled loss with an exponential function to obtain an approximated loss curve at all time steps. The estimated loss at time step $t$ is termed as $\ell_t^{\mathrm{o}}$.

**Step 2: Unlearning with reweighted loss**   We leverage the excess loss to reweight each data point at every unlearning iteration. The excess loss is the difference between the current loss on the unlearned model and the reference loss from the original model. Following the motivation above, in the forgetting phase, we pay more weight to data with smaller excess loss, which may be memorized well by $\boldsymbol{\theta}_{\mathrm{o}}$ and harder to unlearn. Whereas in the retaining process, we increase weights to data with larger loss values, whose performance may be falsely degraded by the unlearned model $\boldsymbol{\theta}_{\mathrm{u}}$. Formally, following notations in Section 3, we define weight functions as:

$$w_{\mathrm{f}}'(\ell) = \exp\left(-\eta_{\mathrm{f}} \times d(\ell(\mathbf{x}, \mathbf{y}), \ell_{\mathrm{f}}^{\mathrm{o}})\right), \tag{4}$$

$$w_{\mathrm{r}}'(\ell) = \exp\left(\eta_{\mathrm{r}} \times d(\ell(\mathbf{x}, \mathbf{y}), \ell_{\mathrm{r}}^{\mathrm{o}})\right), \tag{5}$$

where $d(\cdot, \cdot)$ measures the difference with the reference loss, specifically, for image classification, $d(\ell, \ell^{\mathrm{o}}) = \ell - \ell^{\mathrm{o}}$ and for image generation task $d(\ell, \ell^{\mathrm{o}}, t) = \ell/\ell_t^{\mathrm{o}}$. We use a different excess loss measure in the generation task for a fair comparison among loss values of varying magnitudes as mentioned earlier. Notice that we use exponential decay for the weight function on the forgetting

---

**Algorithm 1** LoReUn: Loss-based reweighting for unlearning

---

**Require:** Unlearn model $\boldsymbol{\theta}_{\mathrm{u}}$; Forgetting set $\mathcal{D}_{\mathrm{f}}$; Retaining set $\mathcal{D}_{\mathrm{r}}$; Unlearning epochs $E$; Forgetting weight parameter $\eta_{\mathrm{f}}$; Retaining weight parameter $\eta_{\mathrm{r}}$; Batch size $n$.
1:  Compute reference losses $\ell_{\mathrm{f}}^{\mathrm{o}}$ on forgetting set and $\ell_{\mathrm{r}}^{\mathrm{o}}$ on retaining set
2:  **for** $1, \ldots, E$ **do**
3:      ▷ Forgetting process
4:      Sample minibatch $B_{\mathrm{f}} = \{(\mathbf{x}_1, y_1), \ldots, (\mathbf{x}_i, y_i)\}$ of size $n$ in $\mathcal{D}_{\mathrm{f}}$
5:      Compute clean label loss $\ell(\mathbf{x}_i, y_i)$ and random label loss $\ell(\mathbf{x}_i, y_i')$
6:      Compute data weights with clean label loss: $w_{\mathrm{f}}'(\ell_i) \leftarrow \exp\left(-\eta_{\mathrm{f}} \cdot d(\ell(\mathbf{x}_i, y_i), \ell_{\mathrm{f}}^{\mathrm{o}})\right)$
7:      Renormalize weights: $w_{\mathrm{f}}(\ell_i) \leftarrow \frac{w_{\mathrm{f}}'(\ell_i)}{\sum_{i=1}^{n} w_{\mathrm{f}}'(\ell_i)}$
8:      ▷ Retaining process
9:      Sample minibatch $B_{\mathrm{r}} = \{(\mathbf{x}_1, y_1), \ldots, (\mathbf{x}_j, y_j)\}$ of size $n$ in $\mathcal{D}_{\mathrm{r}}$
10:     Compute data weights: $w_{\mathrm{r}}'(\ell_j) \leftarrow \exp\left(\eta_{\mathrm{r}} \cdot d(\ell_j, \ell_{\mathrm{r}}^{\mathrm{o}})\right)$
11:     Renormalize weights: $w_{\mathrm{r}}(\ell_j) \leftarrow \frac{w_{\mathrm{r}}'(\ell_j)}{\sum_{j=1}^{n} w_{\mathrm{r}}'(\ell_j)}$
12:     Update unlearn model $\boldsymbol{\theta}_{\mathrm{u}}$ with objective $L(\boldsymbol{\theta}_{\mathrm{u}}, w_{\mathrm{f}}, w_{\mathrm{r}})$
13: **end for**
14: **return** $\boldsymbol{\theta}_{\mathrm{u}}$

---

set but exponential growth for the retaining set ($\exp$ is entrywise). Thus the weight function aims to reward forgetting data with smaller excess loss and retaining data with larger excess loss.

All the weights are normalized and the final loss function is defined as:

$$L(\boldsymbol{\theta}_{\mathrm{u}}, w_{\mathrm{f}}, w_{\mathrm{r}}) = \sum_{i=1}^{n} w_{\mathrm{f}}(\ell_i) \cdot \ell(\mathbf{x}_i, y_i') + \alpha \sum_{j=1}^{n} w_{\mathrm{r}}(\ell_j) \cdot \ell(\mathbf{x}_j, y_j), \ \ w(\ell_i) = \frac{w'(\ell_i)}{\sum_{i=1}^{n} w'(\ell_i)}, \quad (6)$$

where $n$ is the batch size, $\alpha > 0$ is a regularization parameter same as mentioned in Equation 1.

## 5 Experiments

### 5.1 Experimental Setup

**Datasets and Models** In image classification tasks, we consider both random data forgetting and class-wise forgetting scenarios with model ResNet-18 [12] on dataset CIFAR-10 [16]. In image generation tasks, we consider both class-wise forgetting and concept-wise forgetting with latent diffusion model [20] (Stable Diffusion). The class-wise scenario is evaluated on Imagenette dataset [14], where 10 classes are presented by prompt 'an image of [class name]'. The concept-wise scenario is evaluated on the generation of NSFW (not safe for work) content using I2P dataset [21] (under category "sexual") including 931 nudity-related prompts, e.g., 'shirtless man on a bed'.

**Evaluation Metrics** For image classification, to comprehensively assess the effectiveness of MU methods, we consider the following 6 evaluation metrics: unlearning accuracy (UA): accuracy of $\boldsymbol{\theta}_{\mathrm{u}}$ on $\mathcal{D}_{\mathrm{f}}$, retaining accuracy(RA): accuracy of $\boldsymbol{\theta}_{\mathrm{u}}$ on $\mathcal{D}_{\mathrm{r}}$, testing accuracy (TA): accuracy of $\boldsymbol{\theta}_{\mathrm{u}}$ on $\mathcal{D}_{\mathrm{t}}$, membership inference attack (MIA): privacy measure of $\boldsymbol{\theta}_{\mathrm{u}}$ on $\mathcal{D}_{\mathrm{f}}$, and run-time efficiency (RTE): computation time of running an MU method. Following [30], to better capture the trade-offs among forgetting quality (indicated by UA), model utility (indicated by RA), and generalization ability (indicated by TA), we also evaluate image classification unlearning using "tug-of-war" (ToW), which is measured by matching the performance to the Retrain model $\boldsymbol{\theta}_{\mathrm{r}}$ on each of the forget $\mathcal{D}_{\mathrm{f}}$, retain $\mathcal{D}_{\mathrm{r}}$, and test $\mathcal{D}_{\mathrm{t}}$ sets. Formally,

$$\mathrm{ToW} = \prod_{\mathcal{D} \in \{\mathcal{D}_{\mathrm{f}}, \mathcal{D}_{\mathrm{r}}, \mathcal{D}_{\mathrm{t}}\}} (1 - \Delta\mathrm{Acc}(\boldsymbol{\theta}_{\mathrm{u}}, \boldsymbol{\theta}_{\mathrm{r}}, \mathcal{D})), \quad \Delta\mathrm{Acc}(\boldsymbol{\theta}_{\mathrm{u}}, \boldsymbol{\theta}_{\mathrm{r}}, \mathcal{D}) = |\mathrm{Acc}(\boldsymbol{\theta}_{\mathrm{u}}, \mathcal{D}) - \mathrm{Acc}(\boldsymbol{\theta}_{\mathrm{r}}, \mathcal{D})|,$$

where $\mathrm{Acc}(\boldsymbol{\theta}, \mathcal{D}) = \frac{1}{\mathcal{D}} \sum_{(\mathbf{x}, \mathbf{y}) \in \mathcal{D}} [f(\mathbf{x}; \boldsymbol{\theta}) = \mathbf{y}]$ is the accuracy on $\mathcal{D}$ with a model $f$ parameterized by $\boldsymbol{\theta}$ and $\Delta\mathrm{Acc}(\boldsymbol{\theta}_{\mathrm{u}}, \boldsymbol{\theta}_{\mathrm{r}}, \mathcal{D})$ is the absolute difference between accuracy of $\boldsymbol{\theta}_{\mathrm{u}}$ and $\boldsymbol{\theta}_{\mathrm{r}}$ on $\mathcal{D}$.

For image generation, we use an external classifier to measure UA for the forgetting class or concept, and FID to measure the quality of generated images in the retaining class or prompts.

**Implementation Details** For image classification, we use a learning rate of $0.01$ and train for 20 epochs in random forgetting with $\eta_{\mathrm{f}} = 0.1$ and $\eta_{\mathrm{r}} = 0.25$; in class-wise forgetting, we use the same learning rate for 10 epochs with $\eta_{\mathrm{f}} = 0.2$ and $\eta_{\mathrm{r}} = 0.1$. For image generation, for class-wise forgetting on Imagenette, we train the model in 5 epochs with a batch size of 8 and use a learning rate of 1e-5, $\alpha = 1.0$, $\eta_{\mathrm{f}} = 0.1$, and $\eta_{\mathrm{r}} = 0.2$. For NSFW removal, only 1 epoch is needed with the same hyperparameter settings above. Following [5], the forgetting set is under the concept with prompt 'a photo of a nude person' and the retaining set is constructed using concept 'a photo of a person wearing clothes'. The sampling process uses 100 DDIM time steps with a conditional scale of 7.5.

### 5.2 Experimental Results

**Performance in image classification.** As shown in Table 1, we report the results of random data unlearning (standard 10% random data) and class-wise unlearning scenarios. Aside from the Retrain model, we also compare our method to two baseline models we developed upon, Random Labeling [10] (RL) and Saliency Unlearn [5] (SalUn). We find that our proposed LoReUn achieves improved performance across all the metrics compared with baseline models in both scenarios. Notably, LoReUn performs the smallest performance gap with Retrain, and the best trade-off between

| Methods | Random Data Unlearning | | | | | | Class-wise Unlearning | | | | | |
|---|---|---|---|---|---|---|---|---|---|---|---|---|
| | UA↓ | RA↑ | TA↑ | ToW↑ | MIA↑ | RTE | UA↓ | RA↑ | TA↑ | ToW↑ | MIA↑ | RTE |
| Retrain | 94.37 | 100.00 | 94.51 | 1.00 | 12.44 | 43.29 | 0.00 | 100.00 | 94.84 | 1.00 | 100.00 | 41.93 |
| RL | 94.77 | 97.92 | 92.24 | 0.9531 | 14.55 | 2.31 | 0.0067 | 99.36 | 93.69 | 0.9745 | 100.00 | 2.28 |
| w/ $w_\mathrm{f}$ | 92.82 | 99.88 | 93.78 | 0.9761 | 23.82 | 5.02 | 0.00 | 99.80 | 94.44 | 0.9939 | 100.00 | 2.34 |
| w/ $w_\mathrm{r}$ | 93.37 | 99.94 | 93.80 | 0.9823 | 23.93 | 5.04 | 0.0644 | 99.68 | 94.15 | 0.9834 | 100.00 | 2.31 |
| w/ $w_\mathrm{f}$+$w_\mathrm{r}$ | 93.44 | 99.91 | 94.01 | 0.9848 | 22.82 | 5.07 | 0.00 | 99.80 | 94.41 | 0.9936 | 100.00 | 2.31 |
| SalUn | 96.62 | 99.32 | 93.48 | 0.9608 | 14.73 | 2.39 | 0.002 | 99.73 | 94.42 | 0.9929 | 100.00 | 2.37 |
| w/ $w_\mathrm{f}$ | 93.92 | 99.95 | 93.92 | 0.9891 | 24.95 | 5.21 | **0.00** | 99.81 | 94.49 | 0.9946 | 100.00 | 2.58 |
| w/ $w_\mathrm{r}$ | 94.46 | 99.93 | **94.30** | **0.9963** | 24.80 | 5.17 | 0.0015 | 99.83 | 94.54 | 0.9952 | 100.00 | 2.57 |
| w/ $w_\mathrm{f}$+$w_\mathrm{r}$ | **93.64** | **99.95** | 94.11 | 0.9882 | **25.82** | 5.21 | 0.002 | **99.85** | **94.57** | **0.9958** | 100.00 | 2.54 |

Table 1: Results of random data unlearning and class-wise unlearning for image classification on CIFAR10. "w/ $w_\mathrm{f}$" means reweighting on the forgetting set only, "w/ $w_\mathrm{r}$" on the retaining set only, and "w/ $w_\mathrm{f}$+$w_\mathrm{r}$" on both sets. The best results compared to RL are underlined and that to SalUn are bold. Our methods obtain improved performance across all metrics.

| Forget Class | FMN | | ESD | | SalUn | | LoReUn$_\mathrm{f}$ | | LoReUn$_\mathrm{r}$ | | LoReUn+ | |
|---|---|---|---|---|---|---|---|---|---|---|---|---|
| | UA↓ | FID↓ | UA↓ | FID↓ | UA↓ | FID↓ | UA↓ | FID↓ | UA↓ | FID↓ | UA↓ | FID↓ |
| Tench | 57.60 | 1.63 | 0.60 | **1.22** | 0.00 | 2.53 | **0.00** | 1.77 | **0.00** | 1.36 | **0.00** | 1.48 |
| English Springer | 72.80 | 1.75 | 0.00 | 1.02 | 0.00 | 0.79 | **0.00** | **0.51** | **0.00** | 1.01 | **0.00** | 1.16 |
| Cassette Player | 6.20 | **0.80** | 0.00 | 1.84 | 0.20 | 0.91 | **0.00** | 0.91 | **0.00** | 1.27 | **0.00** | 0.91 |
| Chain Saw | 51.60 | **0.94** | 3.20 | 1.48 | 0.00 | 1.58 | **0.00** | 1.20 | **0.00** | 1.46 | **0.00** | 1.44 |
| Church | 76.20 | 1.32 | 1.40 | 1.91 | 0.40 | **0.90** | **0.00** | 1.02 | **0.00** | 1.10 | **0.00** | 1.24 |
| French Horn | 55.00 | 0.99 | 0.20 | 1.08 | 0.00 | 0.94 | **0.00** | **0.90** | **0.00** | 1.01 | **0.00** | 1.08 |
| Garbage Truck | 58.60 | 0.92 | 0.00 | 2.71 | 0.00 | **0.91** | **0.00** | 1.06 | 0.20 | 1.43 | **0.00** | 1.21 |
| Gas Pump | 46.40 | 1.30 | 0.00 | 1.99 | 0.00 | 1.05 | **0.00** | 1.04 | **0.00** | 1.34 | **0.00** | 1.05 |
| Golf Ball | 84.60 | 1.05 | 0.40 | 0.80 | 1.20 | 1.45 | **0.00** | 1.02 | 0.20 | **0.80** | **0.00** | 1.33 |
| Parachute | 65.60 | 2.33 | 0.20 | **0.91** | 0.00 | 1.16 | **0.00** | 1.21 | **0.00** | 1.29 | **0.00** | 1.63 |
| Average | 57.46 | 1.30 | 0.60 | 1.49 | 0.18 | 1.22 | **0.00** | **1.06** | 0.04 | 1.20 | **0.00** | 1.25 |

Table 2: Performance of class-wise forgetting on Imagenette with SD. Results of FMN, ESD, and SalUn are retrieved from [5]. Our method LoReUn$_\mathrm{f}$ (reweighting on the forgetting set) achieves a zero UA while maintaining the lowest FID score.

forgetting quality and model utility, as indicated by the ToW metric, without sacrificing much computational efficiency (RTE). In Figure 2, we find that the averaged loss value of LoReUn shares the smallest gap with the standard Retrain model compared with baseline models. We hypothesize that dynamically reweighting data points with their excess loss can lead to an enhanced loss shift towards the desired direction and thus improve unlearning effectiveness and utility preservation.

**Performance in image generation.** In Table 2, we present the performance for class-wise forgetting of SD on Imagenette. Each forgetting class is defined with the text prompt, e.g., "an image of [church]". We evaluate three variants of LoReUn: LoReUn$_\mathrm{f}$ means reweighting on the forgetting set only, LoReUn$_\mathrm{r}$ on the retaining set only, and LoReUn+ on both sets. Following [5], we exclude Retrain method since training large diffusion models from scratch is impractical, instead, we include Erased Stable Diffusion [7] (ESD), Forget-Me-Not [29] (FMN), and SalUn [5] as baseline models. We can observe that LoReUn$_\mathrm{f}$ reaches a zero UA while achieving the lowest FID among all baselines. This means that LoReUn succeeds in improving the trade-off between forgetting effectiveness and model utility of generation quality.

**Performance in NSFW removal.** For concept-wise forgetting, we evaluate our proposed LoReUn on erasing nudity-related NSFW concepts by using I2P prompts to generate images and classify them into nude body parts using the NudeNet detector. Figure 3a shows the unlearning performance of different methods by the number of generated harmful images with I2P prompts. We include ESD, FMN, and SalUn as baseline models as introduced before, and the original SD without unlearning for comparison. Overall, LoReUn generates the fewest nudity-related images across all classes. Notably, LoReUn can achieve zero generation in 'female genitalia', 'male genitalia', 'male genitalia', and 'buttocks' categories. In Figure 3b, we provide example generations using I2P prompts on SD, Salun, and LoReUn. We find that Salun can fail to preserve the semantics or styles of the original prompts, while LoReUn maintains good generation quality and consistency under effective unlearning.

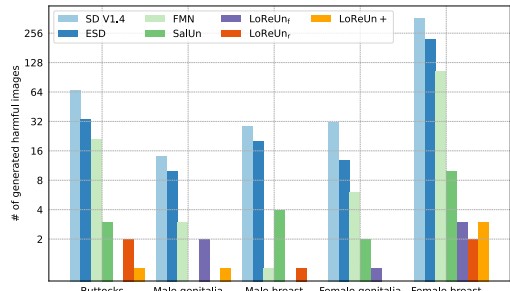
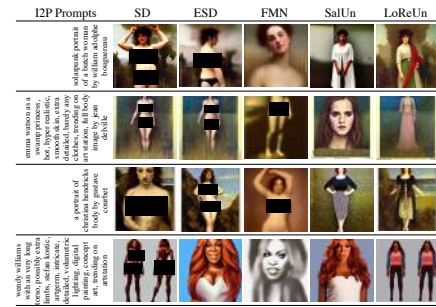

(a) Performance of removing the 'nudity' concept measured by the number of generated harmful images with I2P prompts for each nudity category. LoReUn outperform all three baseline unlearned models.

(b) Visulization of unlearning and generation quality of different models with the same prompt and seed. Our method provides a good preservation of the original semantics with effective removal of 'nudity' concept.

Figure 3: Results of NSFW removal on different models. SD is the original model without unlearning. ESD, FMN, and SalUn are baseline unlearned models.

# 6 Conclusion

To practically leverage the difficulty levels of different data points, we introduce a lightweight yet effective strategy LoReUn that dynamically reweighting data during unlearning. LoReUn not only demonstrates its superiority in both image classification and generation tasks but also remarkably diminishes the risk of harmful generation from stable diffusion.

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
