# OpenReview forum: "LoReUn: Data Itself Implicitly Provides Cues to Improve Machine Unlearning"
_NeurIPS.cc/2024/Workshop/SafeGenAi — SafeGenAi Poster_

### Official Review · Reviewer_bd5i · 2024-10-09
**Very impressive work**

**Rating:** 9
**Confidence:** 4

**Review:**

### Summary
The paper introduces a novel method called **Loss-based Reweighting Unlearning (LoReUn)**, designed to improve machine unlearning (MU) by dynamically reweighting data points based on their difficulty levels. LoReUn uses the loss of each data point on the unlearned model compared to a reference loss from the original model to guide the unlearning process. This approach aims to narrow the performance gap between approximate and exact MU methods while maintaining computational efficiency. The method is evaluated on both image classification and generation tasks, showing improved results compared to existing baselines.

### Strengths
1. **Innovative Approach**: Unlike traditional gradient-based unlearning methods that focus on modifying model weights directly based on gradient information, LoReUn introduces a shift in perspective by using data point loss values to guide the unlearning process. This approach implicitly captures the difficulty level of each data point without requiring costly gradient computations or additional inference steps. This not only simplifies the unlearning strategy but also ensures that data points contributing to undesired memorization are dynamically reweighted according to their impact, making the unlearning process more efficient and effective.

2. **Plug-and-Play Strategy**: LoReUn is presented as a lightweight, plug-and-play solution that can be integrated into existing unlearning methods with minimal computational overhead. This makes it practical for large-scale applications.

3. **Dynamic Reweighting**: The dynamic adjustment of data point weights during the unlearning process helps maintain the balance between forgetting and retaining data, which is crucial for preserving model utility while ensuring effective unlearning.

4. **Comprehensive Evaluation**: The method is evaluated on multiple tasks, including image classification and text-to-image generation, demonstrating its versatility and effectiveness in various scenarios.

5. **Reduction in Computational Cost**: By avoiding the need for explicit evaluation of data point difficulty through gradient-based methods, LoReUn significantly reduces the computational burden compared to traditional approaches that rely on retraining from scratch or gradient computations.

6. **Well Validation**: The results indicate that LoReUn achieves a smaller gap with exact unlearning methods (retraining) compared to baselines, proving its efficiency in minimizing the performance trade-offs between unlearning and model utility.

### Weaknesses
1. **Potential for Optimization**: The use of exponential decay and growth functions for weight adjustment seems somewhat heuristic. More rigorous mathematical justification or exploration of alternative weight functions could improve the method's robustness.

2. **Comparison with Diverse Baselines**: The baselines selected for comparison, primarily Random Labeling and SalUn, might not cover the full spectrum of state-of-the-art unlearning methods. Including a broader range of baselines could strengthen the evaluation.

3. **Dependency on Loss Metrics**: The reliance on the difference between the current and reference loss to guide unlearning might lead to suboptimal results if the loss function does not adequately capture the data's memorization level or if the model's convergence properties vary significantly.

4. **A little bit subjectivity**: The use of exponential decay and growth functions for weight adjustment seems somewhat heuristic. In my opinion, maybe it would be better if there was some sort of mathematical explanation here and why you chose that.

### Recommendation
The proposed LoReUn method presents a compelling approach to improving machine unlearning by leveraging data loss values for dynamic reweighting. Unlike traditional gradient-based methods, LoReUn's strategy simplifies the process and reduces computational overhead, making it a practical alternative for real-world applications. I suggest "strong accept"

---

### Official Review · Reviewer_3TNn · 2024-10-09
**Good problem formulation but not very challenging in terms of problem setup.**

**Rating:** 6
**Confidence:** 3

**Review:**

The paper introduces an unlearning method that, based on data difficulty, reweighs the loss for efficient forgetting. However, it assumes access to the retaining dataset, which might not be available due to privacy concerns. Overall, the method was easy to follow and intuitive.

Some concerns are:
1. What changes would be made if the retaining data subset is unavailable? Ex. in stricter unlearning scenarios.
2. In Fig 2, it is not clear what dataset was used for the plot and conclusion.
3. Additionally, what makes a data point "difficult"? How to classify data points to be difficult? From the experiments, the datasets are pretty standard in computer vision tasks and all images of a dataset are from the same distribution.
4. Ablation results on Eq 6 are missing.
5. In Table 1, what classes are forgotten in "class-wise unlearning"? Is it an average?
6. The paper needs a good amount of rewriting, per se. Some texts are all over the place.

While the proposed method was effective, it is marginally acceptable but not very realistic.

---

### Official Review · Reviewer_o1mi · 2024-10-12
**novel method for machine unlearning**

**Rating:** 8
**Confidence:** 5

**Review:**

This paper proposes a novel strategy for machine unlearning in diffusion models, proposing a novel plug-and-play loss function.
Overall, the proposed method is simple, but seems to be quite effective, comparing to state-of-the-art methods (like SalUn).

The paper is well-written, the experimental setup is comprehensive and shows clear advantages of the proposing method. LoReUn achieves good performance on a set of experimental tasks, which is an obvious strength of this paper. Finally, I think that this work should be interesting for the SafeGenAi workshop audience.